# Evolution of Theories on Doxorubicin-Induced Late Cardiotoxicity-Role of Topoisomerase

**DOI:** 10.3390/ijms252413567

**Published:** 2024-12-18

**Authors:** Jaroslaw Szponar, Erwin Ciechanski, Magda Ciechanska, Jaroslaw Dudka, Sławomir Mandziuk

**Affiliations:** 1Toxicology Clinic, Faculty of Medicine, Medical University of Lublin, Krasnicka 100, 20-718 Lublin, Poland; jaroslaw.szponar@umlub.pl; 2Clinical Department of Toxicology and Cardiology, Regional Specialist Hospital, Krasnicka 100, 20-718 Lublin, Poland; 3Department of Cardiology, Regional Specialist Hospital, Krasnicka 100, 20-718 Lublin, Poland; 4Department of Pulmonary Diseases and Children Rheumatology, Medical University of Lublin, Antoniego Gebali 6, 20-093 Lublin, Poland; 5Department of Toxicology, Medical University of Lublin, Jaczewskiego 8b, 20-090 Lublin, Poland; jaroslaw.dudka@umlub.pl; 6Department of Pneumology, Oncology and Allergology, Medical University of Lublin, Jaczewskiego 8, 20-090 Lublin, Poland; slawomir.mandziuk@umlub.pl

**Keywords:** anthracycline, cardiotoxicity, topoisomerase, dexrazoxane, doxorubicin, cardiac molecular metabolism

## Abstract

Doxorubicin (DOX) has been widely used as a cytotoxic chemotherapeutic. However, DOX has a number of side effects, such as myelotoxicity or gonadotoxicity, the most dangerous of which is cardiotoxicity. Cardiotoxicity can manifest as cardiac arrhythmias, myocarditis, and pericarditis; life-threatening late cardiotoxicity can result in heart failure months or years after the completion of chemotherapy. The development of late cardiomyopathy is not yet fully understood. The most important question is how DOX reprograms the cardiomyocyte, after which DOX is excreted from the body, initially without symptoms. However, clinically overt cardiomyopathy develops over the following months and years. Since the 1980s, DOX-induced disorders in cardiomyocytes have been thought to be related to oxidative stress and dependent on the Fe/reactive oxygen species (ROS) mechanism. That line of evidence was supported by dexrazoxane (DEX) protection, the only Food and Drug Administration (FDA)-approved drug for preventing DOX-induced cardiomyopathy, which complexes iron. Thus, the hypothesis related to Fe/ROS provides a plausible explanation for the induction of the development of late cardiomyopathy via DOX. However, in subsequent studies, DEX was used to identify another important mechanism in DOX-induced cardiomyopathy that is related to topoisomerase 2β (Top2β). Does the Top2β hypothesis explain the mechanisms of the development of DOX-dependent late heart failure? Several of these mechanisms have been identified to date, proving the involvement of Top2β in the regulation of the redox balance, including oxidative stress. Thus, the development of late cardiomyopathy can be explained based on mechanisms related to Top2β. In this review, we highlight free radical theory, iron imbalance, calcium overload, and finally, a theory based on Top2β.

## 1. Introduction

Doxorubicin (DOX) and other anthracyclines (ANTs), including epirubicin, daunorubicin, and idarubicin, have been widely used as anticancer chemotherapeutics in clinical practice for over 50 years [1]. ANT-based treatment regimens are used for small-cell lung cancers, lymphomas, breast cancers, Ewing sarcomas, urinary bladder carcinomas, esophageal carcinomas, stomach carcinomas, hepatocellular carcinomas, and various leukemias in both pediatric and adolescent patients [2,3].

The anticancer mechanism of DOX is associated with its reversible binding to deoxyribonucleic acid (DNA) through intercalation, that is, penetration between DNA strands, and binding to these DNA strands using hydrogen bonds and van der Waals forces [4,5]. DOX can also covalently bind to DNA through a formaldehyde group [6]. Another anticancer mechanism of DOX is related to the stabilization of topoisomerase IIα (Top2α), which is an isoenzyme found in the largest amounts in intensively proliferating cells. Top2 catalyzes changes in the topological state of DNA, interconverting relaxed and supercoiled forms [7,8], causing DNA strand breaks as well as preventing DNA replication and ribonucleic acid (RNA) transcription. The generation of free radicals may also be involved in the anticancer effect of DOX [9]. However, research is lacking on the degree of participation of the above-mentioned mechanisms in various types of cancer.

ANTs produce dose-dependent organ toxicity, especially cardiotoxicity, despite their high efficacy. This manifests as dilated cardiomyopathy (DCM) with systolic and diastolic heart failure, cardiac arrhythmias, ischemic heart disease, or myocarditis and pericarditis [10,11,12,13]. DCM is irreversible because it remodels the heart muscle, causing heart dilatation and reducing the left ventricular ejection fraction (LVEF) (Figure 1). 

DCM is life-threatening and cannot be pharmacologically treated. The risk of developing late heart failure rapidly increases when the cumulative ANT dose is exceeded: 550 mg/m^2^ for DOX (450 mg/m^2^ in cases receiving radiotherapy or with the coexistence of other risk factors), 600 mg/m^2^ for daunorubicin, 1000 mg/m^2^ for epirubicin, 1900 mg/m^2^ for esorubicin, 2–3000 mg/m^2^ for aclarubicin, and 160 mg/m^2^ for mitoxantrone [14,15]. DOX is the ANT most commonly used in the clinic. The incidence rate of dilated cardiomyopathy depends primarily on the DOX dose. The DCM incidence rate is over 35% at a cumulative dose of 650 mg/m^2^. Reducing the cumulative dose to 400 mg/m^2^ reduces the risk of cardiomyopathy to approximately 5%, but even a cumulative dose of 200–360 mg/m^2^ does not completely eliminate the risk of cardiomyopathy [11,16,17,18,19].

Late heart failure may also develop due to the “collateral” cytotoxic effect of DOX on cardiomyocytes [20]. The main cardiotoxic effect of DOX is attributed to oxidative stress and ROS generation. However, an important role of Top2 inhibition in the development of DOX-dependent cardiotoxicity has been recently been proposed. Its cytotoxicity depends in part on the inhibition of Top2α in intensively proliferating cells (e.g., cancer). Additionally, the inhibition of Top2β, which is the only Top2 isoenzyme present in cardiomyocytes, plays a crucial role in cytotoxicity in postmitotic cells, which include cardiomyocytes [21].

In the 1980s, the theory of a free radical mechanism underlying late-onset cardiomyopathy seemed valid. This theory combined many mechanisms into a coherent whole [22,23,24,25]. Since the 1990s, studies have supported the protective effects of antioxidant compounds, which confirmed the free radical theory [26,27,28,29,30]. The understanding of the cardioprotective mechanism of dexrazoxane (DEX) is based on the free radical theory. The metabolite structure of DEX is similar to that of ethylenediaminetetraacetic acid (EDTA), which allows DEX to chelate iron ions, preventing the formation of free radicals from the Fenton reaction. Therefore, antioxidants scavenge the free radicals generated in the DOX-dependent redox reaction, and DEX prevents the formation of free radicals. Since approximately 2000, the validity of the free radical theory has been increasingly recognized. DEX, in addition to its iron ion chelation, affects Top2β, preventing the inhibition of this enzyme by DOX in cardiomyocytes, which results in a cytoprotective effect, decreasing the incidences of cardiomyopathy and congestive heart failure [31,32,33]. The free radical theory is attractive because it explains why changes programmed in the cardiomyocyte via DOX are clinically silent and manifest as dilated cardiomyopathy, even after several years. As such, in this study, we attempted to answer the following question: Can the theory of DOX-induced late cardiotoxicity be explained based on a Top2β-related mechanism? 

## 2. Primary Mechanisms Underlying DOX-Induced Late Cardiotoxicity

### 2.1. Free Radical Mechanism

#### 2.1.1. Background

In the 1980s, the primary cause of DOX-induced DCM with heart failure was thought to be the generation of free radicals. The excessive formation of free radicals disturbed the redox balance and led to oxidative stress. Due to its chemical structure, DOX obtains electrons from nicotinamide adenine dinucleotide (NADH) and nicotinamide adenine dinucleotide phosphate (NADPH) in reactions catalyzed by NADPH–cytochrome P-450 reductase, nitric oxide synthase (iNOS), and other enzymes [10]. After obtaining an electron, DOX forms a semiquinone radical, which, spontaneously and without the participation of enzymes, transfers an electron to molecular oxygen, from which a superoxide anion radical (O_2_^•−^) is formed and DOX returns to its parent form. Therefore, DOX acts as a catalyst for an uninhibited reaction that produces superoxide anion radicals in this redox cycle. This reaction is much more efficient when DOX forms a complex with iron [10,34,35]. The increased levels of O_2_^•−^ initiate a series of redox reactions that produce other ROS that are even more reactive and, therefore, more toxic than O_2_^•−^, e.g., HO^•^, as well as reactive molecules that contain nitrogen in addition to oxygen, e.g., peroxynitrite and ONOO^−^. 

The resulting ROS and products of oxidation damage lipids, proteins, and DNA, leading to many negative changes in the cardiomyocytes. The observed changes in the cardiomyocytes after exposure to DOX, such as a redox imbalance, oxidative stress, disturbances in signaling pathways, disturbances in the intracellular balance of calcium and iron, and disturbances in basic energy metabolism, apoptosis, ferroptosis, and necrosis, are at least partly indirectly, if not directly, dependent on the formation of free radicals (Figure 2).

All of these changes induced by DOX in the cardiomyocyte do not need to be described for the purposes of this article. Therefore, we only call out the most important changes that are directly or indirectly dependent on free radical reactions.

#### 2.1.2. DOX-Dependent Redox and Energy Metabolism Disorders

The continuous, cyclical use of NADH and NADPH changes the dynamic balance of NAD/NADH and NADP/NADPH pairs; these nucleotides may affect the dynamics of oxidation–reduction reactions. Mitochondrial NADH is used after the attachment of DOX to cardiolipin, which is a phospholipid located in the inner mitochondrial membrane [36,37]. Thus, mitochondrial NADH, which is used for ATP synthesis, is largely consumed in the DOX redox cycle, as is cardiolipin, leading to disruptions in the electron transport chain (ETC). Concurrently, the peroxidation of cardiolipin occurs in the inner mitochondrial membrane, mediated by cytochrome c oxidase (COX), resulting in a decrease in ATP synthesis [38,39]. Furthermore, cytochrome c is released from the mitochondria into the cytosol, serving as a signal for cell apoptosis [40]. Additionally, phosphocreatine reserves are depleted [41], and the contractility of the heart muscle is reduced.

An increase in the O_2_^•−^ concentration creates the conditions for its reaction with nitric oxide (NO), resulting in the formation of peroxynitrite ONOO^−^, a reactive and cytotoxic free radical [42,43]. In DOX-treated cardiomyocytes, the increase in the activation of the nuclear factor kappa-light-chain-enhancer of activated B cells (NF-κB) signaling pathway leads to the generation of NO by inducible iNOS [44,45]. In turn, iNOS can substantially inhibit the activity of mitochondrial superoxide dismutase (MnSOD) as a result of the nitration of tyrosine 34, which plays a key role in the activity of this enzyme [46]. The inhibition of the transformation of O_2_^•−^ to H_2_O_2_, which is caused by a decrease in the catalytic efficiency of MnSOD, leads to an increase in the O_2_^•−^ concentration to a level enabling a reaction with NO, the product of which is ONOO^−^. This cyclical cascade of changes during the long-term administration of DOX may disturb mitochondrial energy metabolism. Moreover, O_2_^•−^ is toxic mainly because it damages proteins containing iron-sulphur (Fe-S) centers, such as aconitase, succinate dehydrogenase, and NADH-ubiquinone oxidoreductase [47]. DOX impairs the oxidation of long-chain fatty acids in the cardiac mitochondria during the development of oxidative stress, primarily due to reduced pyruvate levels. However, the lactate levels remain unchanged and acetyl coenzyme A (acetyl-CoA) production increases. Furthermore, the activity of glycolytic enzymes is upregulated, whereas fatty acid oxidation is downregulated [48]. As a result, anaerobic metabolism increases, leading to an increase in glucose metabolism, which, in turn, reduces the DOX-induced mobilization of glucose transporter type 4 (GLUT-4)-containing vesicles to the plasma membrane, limiting subsequent glucose uptake [48,49,50].

The relationship between oxidative stress and the concentrations of ferrous cation (Fe^2+^) and ferric cation (Fe^3+^) ions is well understood through the Fenton reaction. The function of iron in the body is almost exclusively related to cellular respiration processes. Iron is accumulated in the cell in three fractions: stable, labile, and free. Although free iron constitutes approximately 15% of the total cellular iron pool, its concentration must be strictly controlled. An increase in the free iron concentration in the cell initiates the Fenton reaction, which produces free oxygen radicals and damaging proteins, lipids, and nucleic acids [51,52]. Moreover, an iron overload triggers ferroptosis, a programmed iron-dependent cell death, which is different from apoptosis, autophagy, and necrosis [53,54]. The accumulation of free iron in cardiomyocytes causes lipid peroxidation and ferroptosis [55,56]. 

Iron overload in the cardiomyocytes is caused by various mechanisms. Briefly, Fe^3+^ bound to transferrin enters the cell via transferrin receptor protein 1 (TfR1), whereas Fe^2+^ mainly enters the cell via divalent metal transporter 1 (DMT1), as well as L- and T-type calcium channels (LTCC and TTCC) [57]. The ferric iron in the cell may be reduced to ferrous iron by six-transmembrane epithelial antigen of prostate 3 (STEAP3) in the endosome [58]. A cardiomyocyte iron overload due to the labile iron pool may also be possible via ferritinophagy [59,60]. The increases in H_2_O_2_ and O_2_^•−^ concentrations caused by DOX lead to the release of an Fe atom from the [4Fe-4S] cluster of the active center of aconitase, resulting in the formation of a protein with the [3Fe-4S] conformation [61]. The alcohol metabolite of ANTs (with a hydroxyl group at carbon 13) has a similar, but considerably stronger, effect on H_2_O_2_ and O_2_^•−^. The consequences of the transformation of [4Fe-4S] into [3Fe-4S] are more complex than the inhibition of the activity of aconitase, the enzyme catalyzing the isomerization reaction of citrate to isocitrate. The presence of the [3Fe-4S] conformation weakens the transcriptional activity of ferritin, causing ferritinophagy, and stabilizes the mRNA for the transferrin receptor. A decrease in the expression of iron regulatory gene and synthesis of human homeostatic iron regulator protein (HFE), leads to increased iron accumulation in the myocardium and increased cardiotoxicity [62,63] (Figure 3). 

Minotti et al. [10,64,65] found that DOX reduces the synthesis of ferritin while increasing transferrin receptor synthesis. This mechanism may be responsible for the increase in the concentration of the unbound fraction of cellular iron and the intensification of the cardiotoxic effects of DOX. This was confirmed by Kotamraju et al. [66], who showed that chelators that have the ability to pass through the cell membrane and an antibody directed against the transferrin receptor prevented the apoptosis induced by DOX. The results of in vitro studies have suggested that the semiquinone form of DOX and O_2_^•−^ forms with the simultaneous release of iron from ferritin [67,68]. A large amount of iron from transferrin was incorporated into ferritin in cardiomyocytes after incubation with transferrin. Then, within a few hours, the iron bound to ferritin was released [69]. This process is probably dependent on the presence of natural reducing compounds, such as vitamin C or cysteine. However, DOX inhibited the release of iron from transferrin when cardiomyocytes were exposed to transferrin in the presence of DOX, despite the increased incorporation of iron into ferritin in the first phase of the experiment. The described mechanisms led to a labile iron overload, causing lipid peroxidation and ferroptosis, which may be important in the cardiotoxicity of ANTs. ANTs alter the iron balance via HO^•^ and ONOO^–^. The results of in vitro studies by Cairo et al. [70] showed that ONOO^–^ removes the iron atom from the [4Fe-4S] catalytic cluster of aconitase, transforming the enzyme into the cluster-free iron-responsive element-binding protein (IRE-BP). Moreover, the ONOO^–^ formed in the presence of DOX can inhibit mitochondrial aconitase, which markedly impairs the respiration occurring in these organelles [71].

Kim et al. [72] found a competitive relationship between calcium (Ca^2+^) and Fe^2+^ ions. Unlike Fe^3+^, Fe^2+^ strongly inhibits the release of calcium through the sarcoplasmic reticulum (SR) channel, which is induced by Ca^2+^ and DOX ions. Divalent iron thus exhibits similar properties to ruthenium red, a model SR calcium channel blocker. Fe^2+^ may have a strong, direct effect on ryanodine receptor 2 (RYR2), which may be an important mechanism in the toxicity that is dependent on the Fe^2+^ ion loads in cardiomyocytes.

#### 2.1.3. DOX-Induced Changes in the Mitochondria of Cardiomyocytes

Why is the heart the most adversely affected organ by DOX? The heart uses more energy than other organs by pumping about seven tons of blood per 24 h. The brain is an organ that also consumes large amounts of energy, but DOX does not cross the blood–brain barrier, which prevents the mitochondria localized in the central nervous system from being affected [73,74]. The mitochondrial density of the cardiomyocytes is the highest among the mammalian organs: the mitochondria occupy 30% of the cardiomyocyte volume [75]. This strong dependence of cardiomyocyte function on mitochondria means that mitochondrial disorders strongly impact the functioning of the heart. In addition, over 90% of energy (ATP) is produced by mitochondrial respiration. Therefore, any disturbance in the structure or function of the mitochondria affects cardiomyocyte function [75]. 

Mitochondria have long been assigned a key role in DOX-dependent cardiotoxicity. A number of observations have supported this hypothesis. For instance, the enzymatic antioxidant defense in the heart is considerably weaker than that in the liver, with the catalase, superoxide dismutase, and peroxidase activities in the cardiac muscle being at only 10–20% of the levels observed in the liver [10]. This increases the sensitivity of the cardiomyocytes to free radical damage induced by DOX. Additionally, mitochondrial DNA lacks the protective histone proteins found in nuclear DNA, which form a barrier against oxidative damage. Mitochondria lack robust DNA repair mechanisms, unlike nuclear DNA [76].

DOX-induced cardiotoxicity causes alterations in the mitochondrial structure at multiple levels. These changes begin at the biochemical level, with a decrease in cytochrome c oxidase activity, followed by molecular-level changes, such as a reduction in mRNA encoding the COX II subunit, and ultimately at the genetic level, where there is a decrease in the mitochondrial DNA copy number and the increased oxidation of mtDNA [76,77]. DOX damages the energy metabolism in the heart muscle and cardiomyocytes [78,79,80], and the activation of molecular processes leads to the fragmentation of mitochondria [81]. Mitochondrial dysfunction may result from changes in the metabolic pathways of energy production or determine such changes; as such, mitochondrial dysfunction may be both a cause and an effect of changes in the metabolic pathways of energy production. The concept of “metabolic remodeling” indicates that the emerging disturbances in cardiac performance result from a developing energy deficit [82]. In other words, the preferences for obtaining energy from fatty acids to glucose change in the myocardium undergoing hypertrophy [83]. Additionally, a role of energy metabolism disorders has been found in ANT cardiomyopathy [84]. 

#### 2.1.4. Cardiomyocyte Apoptosis

An increase in the ROS concentration can lead to apoptosis [85]. Mitochondria-derived reactive oxygen species play a key role in stimulating the DOX-induced apoptosis pathways in the Fas ligand (FASLG) expressing cardiomyocytes via the nuclear factor of activated T-cells (NFAT) signaling mechanism [86]. DOX stimulates the intrinsic and extrinsic apoptosis pathways [87]. Both pathways are involved in cardiomyocyte apoptosis due to oxidative and antioxidant imbalances [88]. DOX administration leads to the activation of heat shock factor 1 (HSF-1) and induces heat shock protein 25 (HSP-25). Thus, stabilizing the tumor protein P53 (TP53) leads to the generation of proapoptotic factors and the programmed death of cardiac muscle cells [88]. The HSP-70 and HSP-27 expression increases after DOX administration, causing inflammation and affecting the Toll-like-2 receptor (TLR2) signaling pathway, which is involved in apoptosis [89]. Moreover, DOX downregulates protein B kinase (AKT1) and induces caspase 3, which activates apoptosis [90,91]. DOX also enhances the appearance of death receptors (DRs), tumor necrosis factor receptor 1 (TNFRSF1A), death receptor 5 (NFRSF10B)**,** and death receptor 4 (TNFRSF10A) at both the mRNA and protein levels [92]. In addition, TLR2 acts as a novel death receptor through the apoptotic factor caspase 8 and the Fas-associated death domain (FADD), leading to programmed cell death [93]. DOX can affect the expression of the mitochondrial transcription factor GATA-4 (GATA-4) gene, causing the inhibition of mitochondrial synthesis and metabolism and initiating apoptosis [94]. 

#### 2.1.5. Disturbances in the Regulation of Intracellular Calcium Flow

The mechanism of cellular calcium regulation in cardiomyocytes may be ROS-derrypendent, and the free radical mechanism initiated by DOX may play a role in cardiac contractility. The first studies suggesting a relationship between ROS and the Ca^2+^ balance were conducted in the early 1980s [95,96,97]. They found that free radicals were responsible for inhibiting Ca^2+^ uptake in isolated reservoirs of the sarcoplasmic reticulum [95] and dog heart homogenates [96]. Kukreja and Hess [98] and Opie [99] reported that free oxygen radicals affect the SR and initiate the release of Ca^2+^ into the cytosol, which leads to functional changes and cardiomyocyte damage. According to Kawakami and Okabe [100], superoxide anions cause the intensive release of Ca^2+^ from SR reservoirs by affecting the ryanodine receptor of the calcium channel via calmodulin. This effect may be the result of the reaction of the superoxide anion with the sulfhydryl (-SH) groups of ryanodine receptor protein 2 [101]. This hypothesis supports the opening of Ca^2+^ channels by agents that oxidize the -SH group being reversed using compounds that reduce these groups, e.g., cysteine [102]. Meissner [103,104] and Pessah et al. [105] have also suggested the dependence of RYR2 activation on the concentration of free radicals. Calcium-dependent ATPase can be regulated by the ROS located in the endoplasmic reticulum of cardiomyocytes [106]. Moreover, an increase in the concentration of endogenous NO weakens the sensitivity of myofilaments to Ca^2+^, which is accompanied by a weakening of cardiac contractility [107].

Doxorubicin disturbs the Ca^2+^ balance through not only ROS, but also its metabolite. The DOX hydroxyl metabolite (DOX-ol) changes the mitochondrial calcium balance [108,109,110,111]. Mitochondrial calcium imbalances are the result of the selective activation of a calcium channel (cyclosporine A-sensitive) that releases the calcium from mitochondria [109]. Inhibitors of the calcium ion uptake by mitochondria, ruthenium red and cyclosporine A, prevent the irreversible damage to cardiomyocytes induced by DOX [110]. This indicates a direct relationship between mitochondrial calcium balance disturbances and DOX cardiotoxicity [86,112]. 

DOX-ol inhibits sodium–calcium channels, which play an important role in regulating heart contraction. This effect is expressed by increasing the activity of the L-type calcium channel by the sarcoplasmic reticulum calcium pump [113]. Additionally, DOX modifies the calcium exchange regulatory gene, which affects the ryanodine receptor and sarco/endoplasmic reticulum ATPase, SERCA2a (ATP2A2), causing cardiac contractile dysfunction. Intracellular Ca^2+^ accumulation, especially in the sarcoplasmic reticulum of the heart muscle, activates calpains, which are calcium proteases, additionally leading to cell death [114]. Once activated, calpains lead to the degradation of titin, which is the largest protein and the main component of the cardiac sarcomere [113]. Additionally, the increase in the intracytoplasmic calcium concentration leads to the activation of calmodulin-activated protein kinase II (CaMKII) and phospholamban, which increases the apoptosis of mitochondria and, consequently, of cardiomyocytes [115]. The DOX-dependent adenine nucleotide translocase suppression alters the permeability of the calcium-dependent mitochondrial transition pore, causing the irreversible loss of myocardial function in patients treated with DOX [112]. 

To summarize, experimental studies from the 1980s and 1990s showed that intracellular calcium flow is regulated by the influence of ANTs, which may be caused by redox disorders. In the presence of DOX in cardiomyocytes, the influx of calcium ions through free channels increases, the activity of adenylyl cyclase is disturbed, the exchange of Na^+^ and Ca^2+^ ions is inhibited, and the activity of natrium/potassium (Na^+^)/(K^+^) ATPase is decreased. The consequence of these processes is the overload of myocytes with calcium ions and decreases in ATP levels, which are reflected in the impairment of the systolic and diastolic functions of the heart muscle. Hence, the concepts of the cardioprotective effect of calcium channel blockers and β-adrenergic receptor blockers were developed; however, later observations did not confirm these suggestions.

### 2.2. Mechanisms of Topoisomerase 2β in DOX-Induced Dilated Cardiomyopathy

#### 2.2.1. Topoisomerase 2—Background

Evidence of the role of Top2β in ANT cardiomyopathy has been increasing [12,35,116]. Top2 is an enzyme involved in the relaxation of DNA supercoils and the unlinking of DNA linkages during cellular processes, such as DNA replication, transcription, and recombination [117]. DNA topoisomerases are divided into two classes based on their structure and mechanisms [21]. Monomeric topoisomerase type I (Top1) causes DNA single-strand breaks (SSBs) by relaxing supercoiled DNA during the catalytic cycle, while dimeric type Top2 causes double-strand breaks (DSBs) in the DNA [118,119,120]. Topoisomerase 2 has two isoforms in human cells: The first is Top2α, which is present in rapidly proliferating cells, such as tumors and normal intensive dividing cells; thus, Top2α is considered a marker of cell proliferation [121,122], with a high expression in the S and G2 cell cycle phases [21]. The second isoform of topoisomerase, Top2β, is present in all other tissues [12], but only Top2β is present in cardiomyocytes [123]. 

#### 2.2.2. Effect of DOX on Topoisomerase 2

DOX and other ANTs have complex effects on Top2. DOX inhibits the catalytic activity of Top2 through two different pathways. At low concentrations, DOX stabilizes the Top2–DNA complex and blocks DNA regulation [8,124]. At high concentrations, DOX acts as an intercalating agent, preventing Top2–DNA complex formation [4,5]. These effects lead to DNA damage and, finally, to cell death. ANTs inhibit Top2α and Top2β isoforms, causing cardiomyocyte toxicity in addition to their antitumor effects. In the absence of Top2, DOX alone cannot produce DNA double-strand breaks [117,124]. DOX-induced DNA double-strand breaks were found to be dependent on Top2β [12,116]. Lyu et al. [116] found that cardiomyocyte-specific, Top2β-deleted mice pretreated with tamoxifen and DOX shown almost 60% reduction in DNA double-strand breaks found with post mortem immunostaining of the heart. This strongly suggests that Top2β is a target of DOX-induced DNA damage in cardiomyocytes. 

## 3. Theories Explaining Late Cardiotoxicity

### 3.1. Free Radical Theory of DOX-Induced Heart Failure

The vast majority of experimental studies pointing to the free radical mechanism as the primary cause of DOX cardiotoxicity were conducted when DOX was still in the cardiomyocytes and could generate free radicals in the cyclic redox reaction. Relatively few experimental studies have focused on a model in which late cardiomyopathy develops after a long time (weeks or months) after the end of DOX administration, mimicking the clinical realities in humans [125]. The first study with a long follow-up was published in 2003 [126]. Practically no studies have simultaneously analyzed the course of cardiac remodeling in time sequences and the development of overt cardiomyopathy at the functional, morphological, biochemical, and molecular levels. In the absence of such comprehensive data, explaining why clinically silent late cardiomyopathy develops long after the discontinuation of DOX administration is difficult. How does exposure to DOX cause clinically overt cardiomyopathy that occurs several years later, in the absence of DOX in the body?

The theory based on the formation of free radicals as the primary cause of the development of late cardiomyopathy seems plausible due to its substantive consistency. The theory assumes that ANTs transfer an electron from cellular NAD(P)H resources to molecular oxygen as a catalyst, so one ANT molecule can create many superoxide anion radicals, disturbing the redox balance and leading to oxidative stress [127,128,129]. However, the development of heart failure many years after the end of therapy and the absence of DOX in the body requires an explanation. The primary consequence of oxidative stress caused by DOX is mtDNA damage, which causes permanent, constantly increasing mitochondrial electron transport chain dysfunction. Disturbances in the function of the electron transport chain lead to the four-electron reduction of oxygen to water, decreasing in favor of one-, two-, and three-electron reduction, which result in the formation of the reactive oxygen species O_2_^•−^, H_2_O_2_, and HO^•^ [130]. For this reason, free radical damage in the mitochondria constantly continues, increasingly damaging mtDNA and acting as a positive feedback loop, even though DOX is no longer present in the cardiomyocytes. This is a cumulative effect: the more the mtDNA damage, as errors in the formation of mitochondrial electron transport proteins increase, more new free radicals are formed, and new mtDNA damage increases (Figure 2). During this period, the symptoms are clinically silent, but the constant reduction in the efficiency of ATP synthesis is accompanied by cardiac remodeling and, as a consequence, we observe disturbances in heart function in the form of chronic heart failure. The loop theory was first proposed as an explanation for cardiomyopathy that had been clinically hidden for years by Lebrecht et al. [126].

Numerus studies support this hypothesis. Some refer to permanent damage to mtDNA. Zhou et al. deepened our understanding of this problem [112], demonstrating oxidative stress in cardiomyocytes up to five weeks after the last administration of DOX. This suggests a possible involvement of ROS in the cardiotoxicity observed in humans after DOX administration. ANTs induce oxidative damage to mtDNA, which is detectable even many weeks after admission [76]. A mathematical model was proposed by Oliveira and Niederer for the identification of the possible pathways leading to acute and chronic disseminated intravascular coagulation (DIC) [131]. They showed that electron transport chain inhibition may play a major role in the development of acute cardiotoxicity. This is consistent with the findings reported by Pointon et al. [132], where direct mtDNA damage after DOX treatment was responsible for chronic cardiotoxicity, leading to further mitochondrial loss and irreversible dysfunction. Furthermore, Adachi et al. [133] found that chronic ANT cardiotoxicity may be associated with substantial stress-dependent mutations in mtDNA. These results were confirmed by Lebrecht et al. [126], who reported a decrease in the mtDNA/nuclear DNA (nDNA) content in rat hearts. Functional and quantitative mtDNA loss via lesions owing to impaired gene expression may be the reason for the tissue damage occurring many weeks or years after DOX exposure [126,134]. Long-lasting defects in mitochondrial respiration, energy deficits, and sustained oxidative stress are expected, as mtDNA encodes many subunits of the respiratory chain [135]. 

Phosphorus-31 nuclear magnetic resonance (31P NMR) was used by Maslov et al. to detect a 1/3 decline in the myocardial creatine phosphate-to-ATP ratio (PCr/ATP) five weeks after the receipt of a cumulative dose of 25 mg DOX/kg preceding a substantial decline in the LVEF [136]. ANTs impact a mitochondrial enzyme located within its matrix, manganese-dependent superoxide dismutase (SOD2), which is responsible for the conversion of mitochondrial O_2_^•−^ to H_2_O_2_. In addition, Li and Singal [137] found that repeated ANT treatment to achieve a cumulative dose of 15 mg DOX/kg significantly downregulated the expression of SOD2, at both the mRNA and protein levels; its enzymatic activity was decreased for up to three weeks [137,138]. Similar results were reported from a proteomic analysis of long-term daunorubicin-induced cardiotoxicity (cumulative dose of 30 mg DOX/kg) [139]. These results imply that oxidative stress can be substantially compartmentalized in ANT cardiotoxicity because the ANT-induced free radical O_2_^•−^, which is produced within the mitochondrial matrix, does not cross the inner mitochondrial membrane as easily as H_2_O_2_ [135].

ANT-induced oxidative and nitrosative stress may result in a mitochondrial calcium overload [140]. A calcium overload in the mitochondrial matrix affects mitochondrial permeability transition pore (mPTP), leading to the formation of pathological pores in the inner mitochondrial membrane. This results in matrix swelling and membrane rupture, ultimately triggering apoptosis or necrosis. The loss of mitochondrial integrity in this process contributes to ATP depletion and exacerbates oxidative stress [135,141].

In the 1990s, the number of studies on the effectiveness of many antioxidants in various experimental systems sharply increased due to the free radical theory; these studies continue to this day. These studies demonstrated the protective effect of a number of antioxidant factors in the development of congestive heart failure (CHF), such as vitamin C [142]; resveratrol [143]; baicalein biflavonoid [144]; mangiferin, a naturally occurring C-glucosylxanthone [145]; amifostin [146,147]; probucol, an antilipid drug that prevents DOX-induced cardiotoxicity via an antioxidant mechanism [148,149]; α-linoleic acid; melatonin; N-acetylcysteine; and sesame oil through the activation of the antioxidant pathway [150,151,152,153]. Under experimental conditions, the scavenging of the free radicals generated from the redox cycle reactions involving DOX prevents the development of changes leading to cardiotoxicity. The free radical theory, despite its limitations, has long received attention from researchers [35,154]. However, antioxidants that were effective in laboratory tests [12,128,155,156,157] did not produce the expected results in clinical practice [35,158,159], evidencing that oxidative stress is not the only mechanism through which DOX induces cardiotoxicity. While the effective scavenging of ROS alone did not prevent DOX-induced cardiotoxicity, DEX, through its iron chelation properties, confers protection against DOX-induced toxicity in vivo, thereby providing strong support for the iron imbalance hypothesis [160].

### 3.2. How Does DEX Support the Free Radical Theory?

DEX is involved in many aspects of the hypothesis of DOX-dependent late heart failure occurring via oxidative stress. DEX is a DOX-induced cardiotoxicity inhibitor [161], proving its effect in clinical treatment and trials [50,159,162,163]. DEX is the only drug registered by the FDA for this purpose [164]. DEX originates from the family of compounds known as bisdioxopiperazines. DEX reduces ANT-induced toxicity by binding with iron, which prevents the formation of ANT–iron complexes, thus reducing reactive oxygen species formation [165,166]. Unlike its iron-chelating analog ethylenediaminetetraacetic acid, DEX has a ring, not a chain, structure with hydrophilic properties. This is why DEX easily diffuses into cells. In the cell, DEX is converted into a form resembling EDTA, which chelates both free iron and the iron bound in anthracycline complexes, thereby preventing the formation of cardiotoxic ROS [161,167]. Experimental studies have demonstrated that DEX inhibits or counteracts the mechanisms underlying the cardiomyocyte damage induced by DOX. For instance, numerous studies using clinically relevant doses of anthracyclines have found that DEX exerts a cardioprotective effect by inhibiting the apoptotic pathways triggered by anthracycline exposure [85,135,168,169,170]. Sawyer et al. [171] showed that DEX protects neonatal cardiac myocytes from the apoptotic cell death induced by clinically relevant daunorubicin concentrations, but DEX was unable to prevent the necrosis induced by higher doses of daunorubicin. Pretreatment with DEX strongly prevented the oxidation reduction of dichlorofluorescein in the cardiomyocytes caused by relatively high DOX concentrations [172]. 

Another iron chelator, deferiprone, is able to remove excess iron from cardiomyocytes [173]. Deferiprone efficiently binds labile plasma iron in patients with thalassemia [174]. Deferiprone protected an H9c2 cardiomyoblast cell line against oxidative stress induced by tert-butyl hydroperoxide [135,175]. In other in vitro studies, treatment with deferiprone resulted in the inhibition of DOX toxicity in both iron-loaded and normal cardiac cells [135,176]. Despite these results, the number of studies supporting its antioxidant effects remains limited. The inhibition of the DOX-dependent cytotoxicity of cardiomyocytes by iron chelators other than DEX indicates the key role of an iron-related mechanism in DOX cardiotoxicity, regardless of other potential mechanisms.

### 3.3. Theory of Topoisomerase 2β in DOX-Induced Heart Failure

If we reject free radical theory, as suggested by some authors [177,178], we need to explain the long, clinically silent period of the development of cardiomyopathy based on the Top2β mechanism. To date, no separate theory for the development of DOX-dependent cardiomyopathy based on the Top2β mechanism has been proposed. However, recent studies have shown that the mechanism of Top2β inhibition via DOX may have a common denominator with the free radical theory and is associated with the severity of redox disorders and mitochondrial changes. As mitochondrial dysfunction is one of the most important implications of DOX-induced cardiotoxicity mechanisms [78,80,81], we investigated the possible dependence of this dysfunction on Top2β.

The functional and structural changes in mitochondria after treatment with DOX are dependent on the presence of Top2β [31]. The suppression of peroxisome proliferator activation receptor γ coactivator 1-α (PPARGC1A) and peroxisome proliferator-activated receptor γ coactivator 1-β (PPARGC1B) transcription causes mitochondrial dysfunction. PPARGC1A and PPARGC1B regulate the genes involved in the electron transport chain, the tricarboxylic acid cycle, and the β-oxidation of fatty acids via the estrogen-related α receptor (ESR1) and nuclear respiratory factors 1 and 2 (NRF1 and NRF2) [179,180]. Transcriptional coactivators were key regulators in the mitochondria in a heart failure model [179,180]. Zhang et al. [31] found that the transcription of the genes involved in the electron transport chain was not suppressed in mice without Top2β; furthermore, DOX did not cause changes in the oxygen consumption or ultrastructure of Top2β-deficient mice. The mitochondrial mass and density in the cardiomyocytes were reduced in mice without PPARGC1A and PPARGC1B, leading to the rapid development of heart failure. These observations show that the energy reserves in cardiomyocytes are regulated by transcriptional coactivator PGC. Changes in the expression of PGC lead to rapid mitochondrial dysfunction [12]. Thus, disturbances in the mitochondrial electron transport chain via DOX-Top2β-PGC may lead to the one-, two-, and three-electron reduction of oxygen and trigger oxidative stress and the snowball effect in mitochondria, leading to heart failure (Figure 4). 

Another mechanism through which the DOX–Top2β–PGC pathway intensifies the one described above is related to the weakening of antioxidant defenses. DOX alters the gene expression of the antioxidant enzymes superoxide dismutase, peroxiredoxin, and thioredoxin in the presence of Top2β [31]. In this process, PGC-1 is one of the key regulators of superoxide dismutase and plays a major role in ROS-dependent mitochondrial dysfunction [181]. Once PPARGC1A and PPARGC1B expression are suppressed, through either TP53-dependent or -independent mechanisms, the superoxide dismutase expression is also suppressed, promoting ROS production [182,183]. Top2β can directly repress PPARGC1A and PPARGC1B through binding to its promoters in the presence of DOX [31]. The activation of TP53 is increased via its phosphorylation in DOX-induced cardiotoxicity [184]. The inhibition of the TP53 pathway may reduce doxorubicin-induced cardiomyocyte apoptosis [185,186].

Zhang et al. [31] showed that the cardiomyocyte-specific deletion of Top2β prevents DOX-induced DNA DSBs in cardiomyocytes and transcriptome changes, thus affecting mitochondrial biogenesis and preventing ROS formation. Furthermore, the deletion of Top2β protects mice from the development of DOX-induced cardiomyopathy [31]. 

Zhang et al. [31] observed significant increases in the expression of key apoptosis-related genes, such as apoptotic protease activating factor 1 (APAF1), apoptosis regulator BAX (BAX), and fas receptor (FAS), only in the presence of Top2β. These results support the notion that the DOX activation of the TP53 and apoptotic pathways is Top2β-dependent. The absence of Top2β in cardiomyocytes seems to be a cardioprotective factor in DOX-induced cardiotoxicity [12]. Thus, regardless of the many mechanisms through which DOX has cardiotoxic effects via Top2β, the two mentioned mechanisms via the DOX–Top2B–PGC pathway may be related to a cumulative effect, explaining the development of heart failure as an effect of late cardiotoxicity (Figure 3).

### 3.4. How Does DEX Support Top2β Theory?

Strong evidence has been gathered for the protective effect of DEX based on a Top2β-related mechanism in DOX-induced heart failure. Despite the numerous studies demonstrating the relationship between DEX cardioprotection and oxidative stress, some results do not support this interpretation. Only the DEX parent molecule was found to protect neonatal cardiomyocytes from DOX cardiotoxicity, unlike its metabolites or ADR-925 [172]. This indicates that this effect is not dependent on a free radical mechanism. Other more efficient iron chelators, such as deferasirox (ICL670), do not provide cardioprotection against ANT-induced cardiotoxicity [172]. The question arose as to whether this is due to the different bioavailabilities of these compounds. However, deferasirox has an ability to enter cardiomyocytes and efficiently displace Fe^3+^ from its complex with ANT [172]. Despite this, the chelator did not show the same level of cardioprotection against ANT as chemotherapy with DEX. Those observations proved that the ability to chelate iron is not the only determinant of DEX’s cardioprotective effect. Another protective mechanism of DEX in ANT cardiomyopathy was proposed, which is related to the prevention of the ANT inhibition of Top2β, antagonizing the formation of the enzyme cleavage complex and rapidly degrading Top2β [187]. DEX, however, does not induce DSBs of DNA, as do the ANTs [188]. Deng et al. [177] administered DEX to mice with a transiently depleted cardiac topoisomerase isoform, Top2β. Depletion was also observed in H9c2-type cardiomyocytes, but a mutation of the bisdioxopiperazine binding site of Top2β attenuated this effect. Moreover, the accumulation of DOX-induced DNA DSBs by wild-type mice was reduced with DEX, although not in the Top2β mutants. The DEX analog ICRF-161 iron chelator, without the Top2β binding function, did not prevent the accumulation of DOX-induced DSBs [177]. These results support a model of DOX-induced cardiomyopathy caused by Top2β-mediated DSBs and a preventive effect of DEX mediated via Top2β degradation. The mechanisms described in this section support the role of Top2β in the development of DOX-induced cardiotoxicity. However, explaining the development of late cardiomyopathy based on these mechanisms is difficult. This aspect can be explained via the mechanisms related to the DOX–Top2B–PGC pathway described in the previous section.

## 4. Conclusions

Unlike the free radical theory, no hypothesis has been proposed that explains the step-by-step process of the development of DOX-dependent late cardiomyopathy based on the mechanism of action of DOX on Top2β. However, a common denominator in the DOX-dependent free radical generation mechanism and the Top2β-related mechanism can be observed: a disorder in the mitochondrial electron transport chain. Therefore, the mechanism of the positive feedback between ETC disorders and mtDNA damage (“snowball effect”) can be adapted from the free radical theory to a hypothesis explaining the role of Top2β in the development of late cardiotoxicity. If DOX causes disturbances in the ETC via Top2β, the process of ROS formation begins as a result of the one-, two-, and three-electron reduction of oxygen. The generated ROS damage mtDNA, whose electron transport proteins increase the production of ROS, which is facilitated by the weakening of the enzymatic antioxidant defense caused by the DOX–Top2β pathway. The further development of this process at the onset of clinically hidden heart failure can be explained similarly to the classical free radical theory. However, the possibility of explaining the development of late cardiomyopathy based on the DOX–Top2β pathway does not justify abandoning the importance of the classical mechanism directly related to the DOX-dependent production of free radicals.

## Figures and Tables

**Figure 1 ijms-25-13567-f001:**
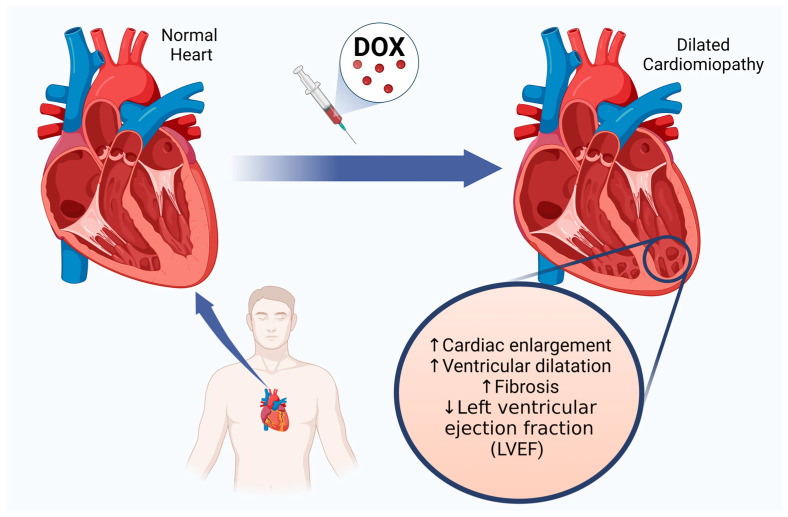
Late cardiotoxic effects of doxorubicin, involving dilated cardiomyopathy caused by cardiac remodeling, ventricle dilatation, progression of fibrosis, and finally, left ventricular ejection fraction reduction; ↑—increase, ↓—decrease.

**Figure 2 ijms-25-13567-f002:**
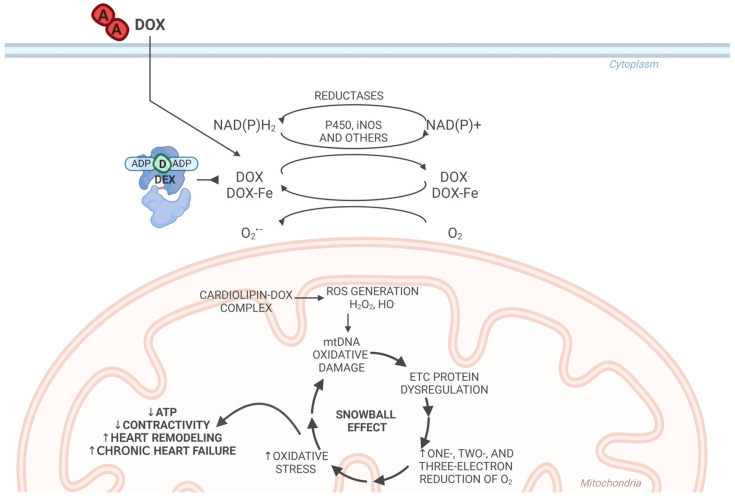
The cytotoxic effect of doxorubicin is directly related to reactive oxygen species (ROS). Briefly, DOX easily obtains electrons from NADH and NADPH in the presence of iron in reactions catalyzed by NADPH, cytochrome P-450 reductase, iNOS, and others. After obtaining an electron, DOX forms a semiquinone radical and transfers an electron to molecular oxygen, forming O_2_^•−^. The presence of O_2_^•−^ favors the generation of other ROS, which damage lipids, proteins, and mitochondrial DNA (mtDNA). NADH is largely consumed in the DOX redox cycle once DOX is attached to cardiolipin in the inner mitochondrial membrane, through which adenosine triphosphate (ATP) synthesis decreases, leading to mitochondrial electron transport chain dysfunction. The four-electron reduction of oxygen to water decreases in favor of one-, two-, and three-electron reduction, which triggers oxidative stress and mtDNA damage. These lead to a positive feedback effect, finally manifesting as heart failure. DEX and the active Top2β cluster together and inhibit DOX–Top2β complex formation, stopping the cycle at an early stage; ↑—increase, ↓—decrease; A—adriamycin (doxorubicin); ADP—adenosine diphosphate; D—dexrazoxane; ETC—electron transport chain.

**Figure 3 ijms-25-13567-f003:**
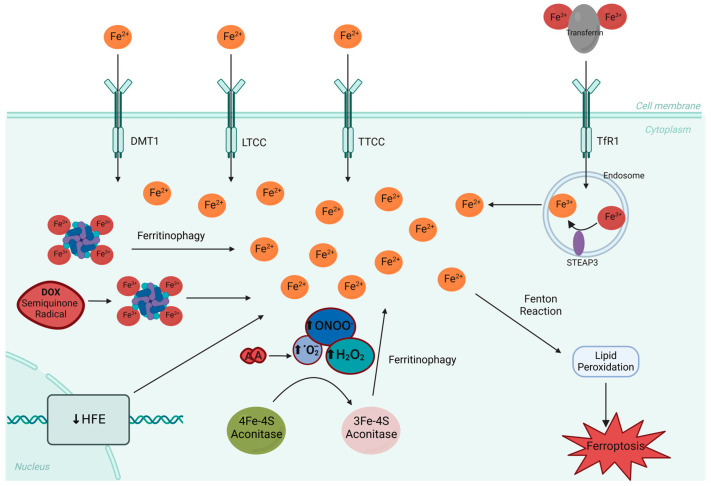
Iron imbalance. Transferrin-bound Fe^3+^ enters the cell through transferrin receptor 1 (TfR1), whereas Fe^2+^ predominantly enters via divalent metal transporter 1 (DMT1), as well as through L- and T-type calcium channels (LTCC and TTCC). Ferric iron is reduced inside the cell’s endosome to ferrous iron by the six-transmembrane epithelial antigen of prostate 3 (STEAP3). The formation of the DOX semiquinone radical concurrently occurs with the release of iron from ferritin. The increased concentrations of H_2_O_2_, O_2_^•−^ and ONOO_−_ induced by DOX trigger the release of an iron atom from [4Fe-4S] aconitase, converting it to the [3Fe-4S] conformation. Consequently, the transcriptional activity of ferritin is reduced, leading to ferritinophagy and decreasing the expression of the iron regulatory gene, lowering the human homeostatic iron regulator protein (HFE) synthesis, in the nucleus. These processes result in the accumulation of a labile iron pool in Fe^2+^ form, which promotes the Fenton reaction, causing lipid peroxidation and triggering ferroptosis, thereby exacerbating cardiotoxicity; ↑—increase, ↓—decrease, DOX—doxorubicin, A—adriamycin (doxorubicin).

**Figure 4 ijms-25-13567-f004:**
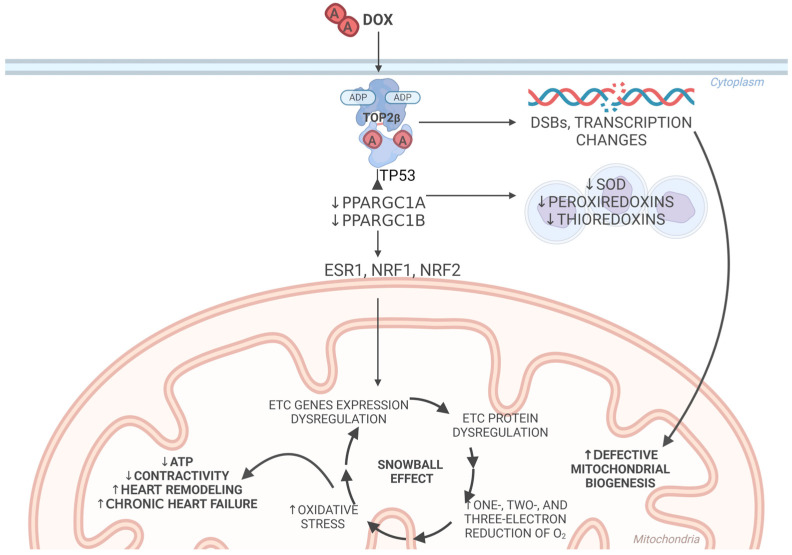
Role of Top2β in the cytotoxic effect of DOX. Once the DOX–Top2β complex is created, DSBs and DNA transcription changes lead to defective mitochondrial biogenesis. Mitochondrial dysfunction is secondary to the suppression of PPARGC1A and PPARGC1B transcription, which regulate genes involved in the electron transport chain, the tricarboxylic acid cycle, and the β-oxidation of fatty acids via ESR1 and NRF1/NRF2. Once the PPARGC1A and PPARGC1B expression is suppressed, the superoxide dismutase (SOD), peroxiredoxin, and thioredoxin expression is also suppressed, promoting ROS production. Thus, disturbances in the mitochondrial electron transport chain lead to the one-, two-, and three-electron reduction of oxygen, triggering oxidative stress and a snowball effect in the mitochondria and leading to heart failure; ↑—increase; ↓—decrease; A—adriamycin (doxorubicin); ADP—adenosine diphosphate; ATP—adenosine triphosphate; DSBs—double-strand breaks; ETC—electron transport chain.

## Data Availability

No new data were created or analyzed in this study.

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
