# Peer review of "Evolution of Theories on Doxorubicin-Induced Late Cardiotoxicity-Role of Topoisomerase"

_ijms, 2024, doi:10.3390/ijms252413567_

Round 1
Reviewer 1 Report
Comments and Suggestions for Authors
The authors are reviewing research related to the development of cardiotoxicity following anthracycline treatment. Two main pathways have been proposed: redox and topoisomerase II-mediated.
The authors need to review the details of how topoisomerase II is thought to contribute to anthracycline-induced cardiotoxicity. The paper is unclear about how this is expected to happen. Some of the dependencies are discussed (such as the impact of the loss of Top2B on cardioxocity), but this section needs to be clarified further.
Comments:
Additional references are needed in several places such as Page 3, lines 87-89, 96-97
Page 8, lines 337 and following: topoisomerase II is not involved in "unwinding of DNA strands" but of "relaxing DNA supercoils and unlinking DNA linkages"
In many places, statements need to be clarified or reworded. The terms used in a number of sentences are incorrect or inappropriate for the meaning intended.
Comments on the Quality of English LanguageMany phrases and sentences need to be revised and clarified. In a number of cases, there are run-on sentences and unclear statements.
Page 2, line 52 - "DNA ruptures" should be "DNA strand breaks"; and DNA and RNA neo synthesis should be "DNA replication" and "RNA transcription".
Page 2, line 56, "organs" should be "organ"
Page 3 - lines 77-80; 90-93 - need to be reworded; unclear statements
Page 6 - lines 231-235, 242-245, 247-251 - these statements need work; they are unclear or misleading in some cases
Page 8, line 353 - "relegation" should be "religation" (as in re-ligation)
Page 10, line 438-439, 455-459 - sentences need rephrasing; these statements are unclear as is
Page 11, lines 468-473; 482-484 - sentences are confusing and need to be reworded to clarify these statements
Page 13, lines 549-551 - statement needs rewording
Author Response
Thank You for Your positive review, we totally agree with the statements. Here are the comments and changes that were made to article:
Comment 1:
"Additional references are needed in several places such as Page 3, lines 87-89, 96-97"
Thank You for pointing this out, we uploaded additional references to prove statements.
Lines 87-89 [22–25]
- Minotti G. Reactions of adriamycin with microsomal iron and lipids. Free Radic Res Commun. 1989;7(3–6):143–8.
- Olson RD, Boerth RC, Gerber JG, Nies AS. Mechanism of adriamycin cardiotoxicity: Evidence for oxidative stress. Life Sci. 1981;29(14):1393–401.
- Doroshow JH. Effect of anthracycline antibiotics on oxygen radical formation in rat heart. Cancer Res. 1983;43(2):460–72.
- Myers CE, Mcguire WP, Liss RH, Ifrim I, Grotzinger K, Young RC. Adriamycin: The role of lipid peroxidation in cardiac toxicity and tumor response. Science (80- ). 1977;197(4299):165–7.
Lines 96-97 [31–33]
- Zhang S, Liu X, Bawa-Khalfe T, Lu L-S, Lyu YL, … Yeh ETH. Identification of the molecular basis of doxorubicin-induced cardiotoxicity. Nat Med. 2012 Nov;18(11):1639–42.
- Keresteš V, Kubeš J, Applová L, Kollárová P, Lenčová-Popelová O, … Jirkovská A. Exploring the effects of topoisomerase II inhibitor XK469 on anthracycline cardiotoxicity and DNA damage. Toxicol Sci. 2024;198(2):288–302.
- Costanzo V, Ratre YK, Andretta E, Acharya R, Bhaskar LVKS, Verma HK. A Comprehensive Review of Cancer Drug–Induced Cardiotoxicity in Blood Cancer Patients: Current Perspectives and Therapeutic Strategies. Curr Treat Options Oncol. 2024;25(4):465–95.
Additional references had been added according to the following guidelines, please check the final manuscript and references.
Comment 2:
"Page 8, lines 337 and following: topoisomerase II is not involved in "unwinding of DNA strands" but of "relaxing DNA supercoils and unlinking DNA linkages" "
"Topoisomerase II an enzyme involved in unwinding of DNA strands "
We agree with that, this mistake is already corrected.
Page 8 lines 346-347
"Top2 is an enzyme involved in the relaxing of DNA supercoils and unlinking DNA linkages"
Comment 3:
Page 2, line 52 - "DNA ruptures" should be "DNA strand breaks"; and DNA and RNA neo synthesis should be "DNA replication" and "RNA transcription".
“This causes DNA ruptures and prevents DNA and RNA neo synthesis”
We agree with that, this mistake is already corrected.
Page 2 line 52,53
“This causes DNA strand breaks and prevents DNA replication and RNA transcription”
Comment 4:
Page 2, line 56, "organs" should be "organ"
“Despite their high effectiveness, ANTs develop dose dependent organs toxicity, especially cardiotoxicity”
Of course, my mistake
Page 2 line 56 and 57
“Despite their high effectiveness, ANTs develop dose dependent organ toxicity, especially cardiotoxicity.”
Comment 5:
Page 3 - lines 77-80; 90-93 - need to be reworded; unclear statements
“The contribution of the above-mentioned cytotoxic anticancer mechanisms is completely reversed in the cytotoxic effect on cardiomyocytes associated with the development of late heart failure [20]. Since the beginning, the main role in the cardiotoxic effect of DOX is attributed to mechanisms initiated by free radical reactions leading to oxidative stress”
We have accordingly revised those statements.
Page 3 lines 76-81
“However despite strong anticancer cytotoxic effect development of late heart failure is also present due to the “collateral” DOX cytotoxic effect on cardiomyocytes [20]. Since the beginning, the main role in the cardiotoxic effect of DOX is attributed to oxidative stress and ROS generation. However, an increasingly important role in the development of DOX-dependent cardiotoxicity had been recently proposed focusing on Top2 inhibition.”
Page 3 – line 90-93
“The cardioprotective effectiveness of dexrazoxane (DEX) was additionally confirmed by the belief that the free radical mechanism was correct, because the DEX metabolite, whose chemical structure was similar to EDTA, chelated iron ions, preventing the formation of free radicals in the Fenton reaction.”
Page 3 lines 89-93
“Moreover, DEX cardioprotective mechanism was based on the free radical mechanism. As we know DEX metabolite’s structure is similar to EDTA which allows it chelating iron ions, preventing the formation of free radicals in the Fenton reaction. Therefore, while antioxidants scavenged free radicals generated in the DOX-dependent redox reaction, DEX prevented the formation of free radicals.”
Comment 6:
Page 6 - lines 231-235, 242-245, 247-251
Thank You for those comments, we rephrased those statements to clarify the crucial meaning.
Page 6 Lines 231-235
The brain is also an organ that consumes a lot of energy, but DOX practically does not cross the blood/brain barrier and does not cause mitochondrial disorders. Cardiomyocytes have the highest mitochondrial density among other organs in mammals - mitochondria occupying 30% of the cardiomyocyte volume [58].
Page 6 lines 236-240
The brain is also an organ that consumes a lot of energy, but DOX practically does not cross the blood/brain barrier thus that it is not affecting mitochondria localized in central nervous system [73,74]. Cardiomyocytes have the highest mitochondrial density among other organs in mammals - mitochondria occupying 30% of the cardiomyocyte volume [75].
Lines 242-245 (*240-244) and 247-251
“For example, in the heart, compared to, for example, the liver, the enzymatic antioxidant defense is much weaker, because the activity of catalase, superoxide dismutase and peroxidase in the heart muscle is at a level of 10-20% of that observed in the liver [10]. This makes the cardiomyocyte more sensitive to the mechanism of free radical damage initiated by DOX. Moreover, mitochondrial DNA does not have the same protection as nuclear DNA, in which histone proteins constitute a barrier that protects nDNA against oxidative damage. Mitochondria do not have repair systems like nDNA [59].
As it was mentioned, from the beginning of DOX-induced cardiotoxicity studies, evidences have been provided for changes in mitochondria, in the ultrastructural picture, changes at the biochemical level - decrease in cytochrome-c oxidase activity, and later at the molecular level - decrease in mRNA coding for the subunit in COX II and at the genetic level - decrease in copy number and oxidation of mtDNA [59,60].”
Page 6 Lines 246-258
For instance, the enzymatic antioxidant defense in the heart is considerably weaker than in the liver, with the activities of catalase, superoxide dismutase, and peroxidase in cardiac muscle functioning at only 10-20% of the levels observed in the liver [10]. This increases the sensitivity of cardiomyocytes to free radical damage induced by DOX. Additionally, mitochondrial DNA lacks the protective histone proteins found in nuclear DNA, which form a barrier against oxidative damage. Unlike nuclear DNA, mitochondria also lack robust DNA repair mechanisms [76].
As previously mentioned, studies on DOX- induced cardiotoxicity have consistently shown alterations in mitochondrial structure at multiple levels. These changes begin at the biochemical level, with a decrease in cytochrome c oxidase activity, followed by molecular-level changes, such as a reduction in mRNA encoding the COX II subunit, and ultimately at the genetic level, where there is a decrease in mitochondrial DNA copy number and increased oxidation of mtDNA. [76,77].
Comment 7:
Page 8, line 353 - "relegation" should be "religation" (as in re-ligation)
Thank You, changes are already made.
Page 8 line 360 religation
Comment 8:
Page 10, line 438-439, 455-459 - sentences need rephrasing; these statements are unclear as is
We agree with those comments, statements are modified.
Page 10 line 438-439
Calcium overload of mitochondrial matrix is proved have impact on mitochondrial permeability transition pore (mPTP), affecting function of inner mitochondrial membrane, what leads to ATP depletion and further oxidative stress damage [119,125].
Page 10 line 441-446
Calcium overload in the mitochondrial matrix has been shown to affect mitochondrial permeability transition, leading to the formation of pathological pores (mPTP) in the inner mitochondrial membrane. This results in matrix swelling and membrane rupture, ultimately triggering apoptosis or necrosis. The loss of mitochondrial integrity in this process contributes to ATP depletion and exacerbates oxidative stress [136,142].
Page 10 line 455-459
However, not only scavenging the formed ROS effectively prevented DOX cardiotoxicity. One of the strongest pieces of supporting evidence for the iron hypothesis is that the iron chelator, DEX is protective against doxorubicin-induced toxicity in vivo [144].
Page 10 line 461-464
However, it was not only the effective scavenging of ROS that prevented DOX-induced cardiotoxicity. Dexrazoxane, through its iron chelation properties, confers protection against DOX-induced toxicity in vivo, thereby providing strong support for the iron imbalance hypothesis [161].
Comment 9:
Page 11, lines 468-473; 482-484 - sentences are confusing and need to be reworded to clarify these statements
Of course, thank You for pointing this out, we corrected those sentences.
Page 11 lines 468-473
Upon, in the cell, DEX is transformed into a form similar to EDTA, a potent iron chelator, causing iron displacement from the ANT-Fe complex formed in the cell [152]. Experimental studies have shown that DEX inhibits or counteracts the mechanisms leading to cardiomyocyte disorders and death induced by DOX. For example, many studies using clinical equivalent doses of ANT demonstrated the cardioprotective effect of DEX with simultaneous inhibition of ANT-induced activation of the apoptotic pathway [68,119,153–155].
Page 11 line 472-481
Unlike its iron-chelating analogue ethylenediaminetetraacetic acid. DEX has not a chains but a rings structure with hydrophilic properties and that is why easily diffuses into cells. In the cell, DEX is converted into a form resembling EDTA, which chelates both free iron and iron bound in anthracycline complexes, thereby preventing the formation of cardiotoxic ROS [162,169]. Experimental studies have demonstrated that DEX inhibits or counteracts the mechanisms underlying cardiomyocyte damage induced by DOX. For instance, numerous studies using clinically relevant doses of anthracyclines have shown that DEX exerts a cardioprotective effect by inhibiting the apoptotic pathways triggered by anthracycline exposure [85,136,170–172].
Page 11, lines 482-484
It has been shown that deferiprone efficiently bind cellular iron, both free and accumulated within the mitochondria and lysosomes [159]. So far limited number of studies have proved the antioxidant potential of deferiprone.
Page 11 lines 488-489 and 492-493
It has been demonstrated that deferiprone efficiently binds labile plasma iron in patients with thalassemia [176].
However, despite those results, the number of studies supporting its antioxidant effects of deferiprone remains limited
Comment 10:
Page 13, lines 549-551 - statement needs rewording
In study conducted by Zhang et al. [164] significant increase in transcripts of important genes of apoptosis pathway (Apaf1, Bax and Fas) was observed only in the presence of Top2β. This additionally proves that activation of p53 and apoptosis pathway after DOX treatment depends on the Top2β enzyme.
Those final comments are crucial, we already rephrased the statements.
Page 13 lines 451-454
In a study conducted by Zhang et al. [31], a significant increase in the expression of key apoptosis-related genes, such as Apaf1, Bax, and Fas, was observed only in the presence of Top2β. Stated results support the notion that DOX activates the p53 and apoptotic pathways in a Top2β -dependent manner

Reviewer 2 Report
Comments and Suggestions for Authors
Dr. Szponar wrote a nice review about the mechanisms of DOX-induced cardiotoxicity. They first discussed the well-known mechanism about oxidative stress in the DOX-induced cardiotoxicity. Then, they expanded the discussion to explore the role of topoisomerase 2β (Top2β) in the DOX-induced cardiotoxicity. It is a well-written manuscript. However, the reviewer dose has some concerns about the current manuscript.
The current manuscript looks targeting small group of audiences who need to have extensive mitochondrial knowledge. Otherwise, the manuscript is hard to be understood by audiences without extensive mitochondrial knowledge. Authors should discuss more about basic concepts and increase the broadness of audiences.
Line 114-115 “It is unusual and dangerous because after giving up an electron, the semiquinone radical returns to the parent DOX form, ready to take another electron”. If the electron is released from the semiquinone radical, the semiquinone should be a radical anymore because it does not have unmatched electron.
Line 149 “ ---caridiolipin---- protein---” Cardiolipin is a phospholipid but not a protein
Line 150 “Thus, the mitochondrial energy fuel NADH for ATP synthesis is largely consumed in the redox cycle of DOX combined with cardiolipin. At the same time, the activity of cytochrome c oxidase decreases and subsequently, ATP—”
It is hard to understand here. Please explain how DOX treatment leads to decreased cytochrome c oxidase activity” due to decreased cardiolipin content or oxidation or subunits damage???
Line 167-168 “During the development of oxidative stress, DOX reduces the oxidation of long-chain fatty acid in cardiac mitochondria and increases glucose metabolism, promoting anaerobic metabolism”
Please give detail here how DOX treatment leads to impaired substrate flexibility. Due to damaged pyruvate dehydrogenase or something else?
2.1.3. Iron imbalance
Please include a scheme figure for this topic. There is a difficulty to following in this section without further illustration.
Author Response
Thank You for Your positive review, we totally agree with the statements. Here are the comments and changes that were made to article:
Comment 1:
Line 114-115 “It is unusual and dangerous because after giving up an electron, the semiquinone radical returns to the parent DOX form, ready to take another electron”. If the electron is released from the semiquinone radical, the semiquinone should be a radical anymore because it does not have unmatched electron.
I agree with that point, here is the revised part.
Page 3 Line 109-112
After taking an electron, DOX forms a semiquinone radical, which spontaneously, without the participation of enzymes, transfers an electron to molecular oxygen, from which a superoxide anion radical (O2•-) is formed and DOX returns to its parent form.
Comment 2:
Page 4 Line 149 “ ---caridiolipin---- protein---” Cardiolipin is a phospholipid but not a protein
Sure, that is already corrected.
Page 4 Line 146 phospholipid
Comment 3:
Line 150 “Thus, the mitochondrial energy fuel NADH for ATP synthesis is largely consumed in the redox cycle of DOX combined with cardiolipin. At the same time, the activity of cytochrome c oxidase decreases and subsequently, ATP—”
It is hard to understand here. Please explain how DOX treatment leads to decreased cytochrome c oxidase activity” due to decreased cardiolipin content or oxidation or subunits damage???
I modified the statements so that this mechanism is clear, moreover I added citations to prove those statements.
Page 4 lines 147-153
Thus, mitochondrial NADH, which is used for ATP synthesis, is largely consumed in the redox cycle of DOX in combination with cardiolipin, leading to disruptions in the ETC. Concurrently, the peroxidation of cardiolipin in the inner mitochondrial membrane, mediated by cytochrome c oxidase (COX), occurs, resulting in a decrease in ATP synthesis [38,39]. Furthermore, the release of cytochrome c from the mitochondria into the cytosol is observed, serving as a signal for cell apoptosis [40].
Comment 4:
Line 167-168 “During the development of oxidative stress, DOX reduces the oxidation of long-chain fatty acid in cardiac mitochondria and increases glucose metabolism, promoting anaerobic metabolism”
Please give detail here how DOX treatment leads to impaired substrate flexibility. Due to damaged pyruvate dehydrogenase or something else?
This mechanism is clearly described in the corrected version of the article.
Page 5 lines 167-174
During the development of oxidative stress, DOX impairs the oxidation of long-chain fatty acids in cardiac mitochondria, primarily due to reduced levels of pyruvate, while lactate levels remain unchanged and Acetyl-CoA production increases. Furthermore, the activity of glycolytic enzymes is upregulated, while fatty acid oxidation is downregulated [48]. As a result, anaerobic metabolism is promoted, leading to an increase in glucose metabolism. This, in turn, reduces DOX-induced mobilization of GLUT-4-containing vesicles to the plasma membrane, thereby limiting subsequent glucose uptake [48–50].
Comment 5:
2.1.3. Iron imbalance
Please include a scheme figure for this topic. There is a difficulty to following in this section without further illustration.
Please find the attached figure describing iron imbalance mechanisms.

Fig. 3 Iron imbalance
Ferric iron (Fe³⁺), bound to transferrin, enters the cell through transferrin receptor 1 (TFR1), while ferrous iron (Fe²⁺) predominantly enters via divalent metal transporter 1 (DMT1), as well as through L-type and T-type calcium channels (LTCC and TTCC). Once inside the cell's endosome, ferric iron is reduced to ferrous iron by six-transmembrane epithelial antigen of prostate 3 (STEAP3). The formation of the DOX semiquinone radical occurs concurrently with the release of iron from ferritin. Additionally, the increased concentrations of H₂O₂, O₂•⁻, and ONOO⁻ induced by DOX trigger the release of an iron atom from the [4Fe-4S] aconitase, resulting in its conversion to the [3Fe-4S] conformation. Consequently, the transcriptional activity of ferritin is reduced, leading to ferritinophagy and a decrease in the expression of the iron regulatory gene, human homeostatic iron regulator protein (HFE), in the nucleus. These processes result in the accumulation of the labile iron pool (Fe²⁺), which promotes the Fenton reaction, causing lipid peroxidation and triggering ferroptosis, thereby exacerbating cardiotoxicity.

Round 2
Reviewer 1 Report
Comments and Suggestions for Authors
Thank you for addressing several of my initial concerns. However, please note that there are still many English errors that need correcting. Your manuscript needs to be read by a native English speaker and corrected for grammar, spelling, and basic errors. This includes errors in the text and the headings that affect the clarity of your submission.
Comments on the Quality of English LanguageThere are still statements throughout the manuscript that need to to be addressed. This needs to be read by an English speaker to correct these issues before moving the manuscript forward.
Author Response
File has been revised with professional MDPI english services. Please see the new attached version.
Round 3
Reviewer 1 Report
Comments and Suggestions for Authors
Thank you for revising the manuscript. The language is much clearer now.